# Efficient conversion of propane in a microchannel reactor at ambient conditions

Chunsong Li [1,4], Haochen Zhang [1,4], Wenxuan Liu [1], Lin Sheng[1], Mu-Jeng Cheng [2], Bingjun Xu [3], Guangsheng Luo[1] ✉ & Qi Lu [1] ✉

The oxidative dehydrogenation of propane, primarily sourced from shale gas, holds promise in meeting the surging global demand for propylene. However, this process necessitates high operating temperatures, which amplifies safety concerns in its application due to the use of mixed propane and oxygen. Moreover, these elevated temperatures may heighten the risk of over-oxidation, leading to carbon dioxide formation. Here we introduce a micro-channel reaction system designed for the oxidative dehydrogenation of propane within an aqueous environment, enabling highly selective and active propylene production at room temperature and ambient pressure with miti-gated safety risks. A propylene selectivity of over 92% and production rate of 19.57 mmol $m_{Cu}^{-2}$ $h^{-1}$ are simultaneously achieved. This exceptional perfor-mance stems from the in situ creation of a highly active, oxygen-containing Cu catalytic surface for propane activation, and the enhanced propane transfer via an enlarged gas-liquid interfacial area and a reduced diffusion path by establishing a gas-liquid Taylor flow using a custom-made T-junction micro-device. This microchannel reaction system offers an appealing approach to accelerate gas-liquid-solid reactions limited by the solubility of gaseous reactant.

Propylene ($C_3H_6$) serves as a crucial feedstock within the petrochem-ical industry, with its primary production methods being steam cracking and fluid catalytic cracking (FCC)[1–3]. However, these con-ventional methods are being stretched by the escalating global demand for propylene[4,5]. The propane ($C_3H_8$) dehydrogenation (PDH) process, capitalizing on the abundant propane in shale gas, has been developed to bridge this supply-demand gap[6,7]. Nevertheless, the PDH process suffers from drawbacks including its endothermic nature requiring substantial energy input, and rapid catalyst deactivation due to coke formation[5,8]. Oxidative dehydrogenation of propane (ODHP) presents an appealing alternative. This process is exothermic, allowing it to operate at lower temperatures and reducing energy requirements[9–11]. Additionally, the presence of oxygen ($O_2$) in ODHP suppresses the coke formation on the catalysts, and thus extending their lifetime[12,13]. A central research theme of ODHP is the development

of highly selective catalytic system for propylene formation while concurrently inhibiting its consecutive overoxidation to form carbon dioxide ($CO_2$). The typical selectivity of propylene in ODHP process using conventional metal oxide catalysts, such as vanadium- and chromium-based materials, is less than 70% at typical experimental conditions (400-500 °C and 1 atm)[14]. Recently, boron-containing materials, including supported boron oxide, hexagonal boron nitride, and boron-doped zeolites, have been reported to selectively produce propylene with suppressed $CO_2$ formation in ODHP process[15–21]. Alkanes are proposed to be activated on boron-based reaction sites via a radical mechanism, which typically requires tem-peratures exceeding 500 °C to achieve substantial rates[22,23]. The high operating temperature fails to take adavantage of the strong thermo-dynamic driving force in ODHP, and enhances the mobility of hydro-xylated boron centers, causing catalyst deactivation[24,25]. Moreover,

[1]State Key Laboratory of Chemical Engineering, Department of Chemical Engineering, Tsinghua University, Beijing, China. [2]Department of Chemistry, National Cheng Kung University, Tainan, Taiwan. [3]College of Chemistry and Molecular Engineering, Peking University, Beijing, China. [4]These authors contributed equally: Chunsong Li, Haochen Zhang. ✉e-mail: gsluo@tsinghua.edu.cn; luqicheme@mail.tsinghua.edu.cn

high temperature reactions employing a mixture of alkane and oxygen carry safety risks, especially at large scale.

The key to addressing the challenges in ODHP lies in the development of efficient reaction system that can achieve high selectivity and activity in propylene production at low temperatures (e.g., <200 °C). We recently identified an aqueous reaction system that uses copper (Cu) powder as the catalyst and $O_2$ as the oxidant, demonstrating the capability to activate light alkanes at room temperature and ambient pressure with appreciable rates[26]. Notably, with a batch reactor, propane can be transformed into propylene with up to 94% selectivity and a production rate of 0.71 mmol $m_{Cu}^{-2}$ $h^{-1}$. However, the inefficient mass transfer of propane due to its low solubility in the aqueous solution (~1.1 mmol $L^{-1}$) hinders the further enhancement of reaction efficiency for practical implementation and complicates kinetic analysis for mechanistic understanding. In addition, the use of $C_3H_8$ and $O_2$ mixture still presents a safety risk in operation.

Herein, we report a microfluidic reaction system with a Cu microtube serving as both the catalyst and the microchannel reactor to simultaneously address the issues of inefficient mass transport and safety risk, achieving a 27-fold increase in reactivity compared to a batch reactor while maintaining the propylene selectivity over 92%. We used a custom-made T-junction microdevice to disperse gaseous reactants ($C_3H_8 + O_2$) into a sulfuric acid ($H_2SO_4$) solution, which formed a gas-liquid Taylor flow prior to entering the Cu microtube. Flow pattern analysis assisted by high-speed camera imaging reveals a substantial enhancement in mass transfer, which can be attributed to the increased gas-liquid interfacial area and a shortened diffusion path for the reactant gas to reach the Cu surface. Importantly, our system also mitigates the potential explosion risk associated with the $C_3H_8 + O_2$ mixture, as the gaseous mixture is segmented into small millimeter-scale portions by plugs of aqueous solution. We conducted kinetic investigations to determine the reaction orders and apparent activation energy. Density functional theory (DFT) calculations were also carried out to explore the reaction pathways leading to the formation of propylene and other minor products (i.e., n/i-propanol). Our microfluidic system demonstrates high selectivity and activity in producing propylene from propane at room temperature and ambient pressure. This approach presents a promising pathway for enhancing reactivity of other gas-liquid-solid multiphase reactions, particularly those hindered by the low solubility of gaseous reactants in a liquid medium, such as $H_2O_2$ synthesis using $H_2$ and $O_2$[27,28], aerobic alcohol oxidation using $O_2$[29-31], and electrochemical conversion of $CO_2$[32-34], $O_2$[35-37], and $N_2$[38,39] on metallic catalyst surfaces.

## Results

### Gas-liquid Taylor flow in a Cu microtube reactor

A commercial Cu microtube with an inner diameter of 550 μm and length of 3 m was used as the tubular microchannel reactor, with its inner surface as catalyst. The morphology and chemical composition of Cu microtube were characterized using scan electron microscopy (SEM) equipped with energy dispersive spectroscopy (EDS) (Supplementary Fig. 1). Prior to entering the Cu microtube reactor, the reactant gases, composed of $C_3H_8$ and $O_2$ with a predetermined ratio, were dispersed into the aqueous $H_2SO_4$ solution using a custom-made capillary embedded step T-junction microdevice[40] (see Supplementary Fig. 2 for the scheme of the experimental setup). This results in the formation of a stable gas-liquid Taylor flow (Fig. 1a), as recorded by a high-speed camera in transparent polypropylene tubes connected to both ends of the Cu microtube with the identical diameter. This Taylor flow pattern in microchannel, containing Taylor bubbles, liquid plugs, and thin liquid film between every Taylor bubbles and inner surface, has been demonstrated to significantly enhance mass transfer characteristic of gas molecules due to the large interfacial area, special circulation flow field and the short path required for molecular diffusion across the liquid film[40-43]. In addition, Taylor flow is characterized

for its uniformity, making it suitable for conducting mass transport characteristic analyses. Other flow patterns such as bubbly flow and annular flow (Supplementary Fig. 3), have limitations in propane activation: the bubbly flow, while increasing the interfacial area, does not reduce the diffusion path, and the annular flow, with an extremely low liquid to gas ratio, does not provide a sufficient liquid supply necessary for propane activation.

The flow patterns as well as the mass transfer characteristics, i.e., thickness of liquid film and gas-liquid interfacial area, were investigated at various liquid flow rates and gas flow rates. Decreasing the liquid flow rates at a fixed total flow rate of $C_3H_8 + O_2$ of 2 mL $min^{-1}$ maintains the pattern of Taylor flow unchanged but elongates the gas bubbles (Fig. 1b), which essentially increases the gas-liquid interfacial area per unit of volume (Fig. 1c) calculated based on the dimensions of the gas and liquid components in Taylor flow (see Supplementary Note 1 for details) and promotes the dissolution of gas molecules to the liquid phase[44,45]. This gas-liquid interfacial area can reach a value of ~8000 $m^2/m^3$, as opposed to only ~100 $m^2/m^3$ for conventional batch reactors used in chemical processes[46], highlighting the enhanced mass transfer in the Cu microtube reactor. The liquid film thickness decreases as the liquid flow rate decreases (Fig. 1d), facilitating the diffusion of dissolved gas molecules to the Cu catalyst surface. The increase of gas-interfacial area and decrease of thickness of liquid film both improve the mass transfer of the gaseous reactants to the Cu surface. We also investigated the mass transfer characteristics at different gas flow rates with a fixed liquid flow rate of 0.2 mL $min^{-1}$. Increasing the gas flow rates enlarges the gas-interfacial area and reduces the thickness of liquid film (Supplementary Fig. 4), thus improving the mass transfer, which is in good agreement with previous studies[44,45,47].

### Propane activation in Cu microtube reactor

The activation of propane was then carried out in this Cu microtube reactor, which also serves as the catalyst, at ambient pressure and room temperature. The mixture of $C_3H_8$ and $O_2$ with a molar ratio of 3:1 was used as the gas feed, and 0.5 M $H_2SO_4$ was chosen as the reaction medium. The Taylor flow was generated as described above. After passing through the Cu microtube, the gas bubbles were collected using downward displacement of water, and then analyzed using a gas chromatograph (GC), while the liquid was collected separately and analyzed using a nuclear magnetic resonance (NMR) spectrometer. Propylene was the main product of propane activation, with a selectivity up to 93% under all conditions. A small amount of methane and ethylene were also detected, which are the common side products in the oxidative dehydrogenation of propane[48-50]. Other minor liquid products, including propionic acid, acetone, i-propanol, acetic acid and formic acid were also detected. No appreciable overoxidation was observed as the combined selectivity of CO and $CO_2$ is less than 1%.

The activation of propane in the Cu microtube was first evaluated at various liquid flow rates with a fixed gas flow rate (Fig. 2a). At a fixed gas flow rate of 2.00 mL $min^{-1}$, we observed a significant increase in propane oxidation rate from 4.55 to 15.97 mmol $m_{Cu}^{-2}$ $h^{-1}$ as the liquid flow rate decreased from 1 to 0.25 mL $min^{-1}$. The increase in propane oxidation rate at this liquid flow rate range can be attributed to the improved mass transfer as revealed by the increased gas-liquid interfacial area and decreased liquid film thickness (Figs. 1c and 1d). However, at the liquid flow rate ranging from 0.25 to 0.1 mL $min^{-1}$, the propane oxidation rate plateaus, suggesting that the reaction was under kinetic control at these conditions and that any further improvement in mass transfer resulting from reducing the liquid flow rate would no longer be advantageous. The surprising decrease of reaction rate at further improved mass transfer is likely due to the decreasing concentration of $H_2SO_4$ and $O_2$ as the Taylor flow passes through the Cu microtube, primarily due to the consumption during Cu dissolution. The reduction in the $H_2SO_4$ concentration does not

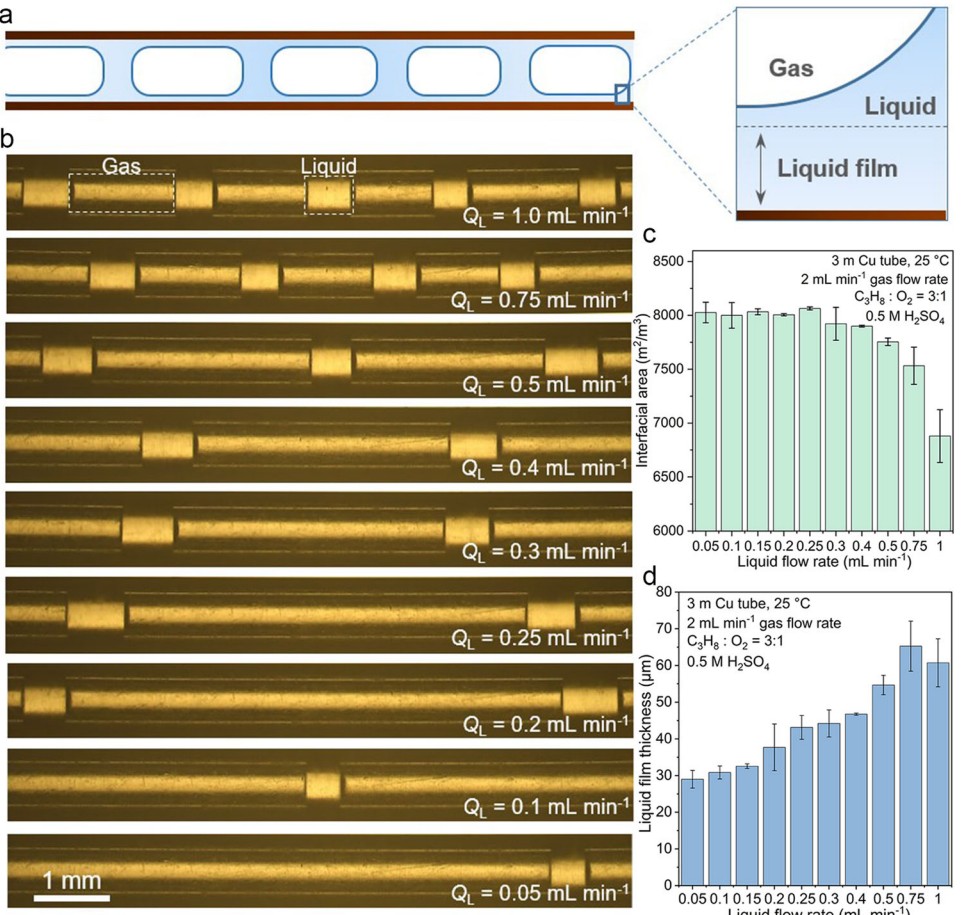

**Fig. 1 | Enhanced mass transfer characteristics of C₃H₈ in the microchannel reactor. a** Schematic of the gas-liquid Taylor flow. **b** High-speed camera photographs of the gas-liquid Taylor flow at various liquid flow rates ($Q_L$) in a transparent polypropylene tube connected to both ends of the Cu microtube with identical diameter. The total gas flow rate ($Q_G$) was fixed at 2 mL min⁻¹. **c** Gas-interfacial area calculated from gas-liquid flow pattern at various liquid flow rates with a fixed gas flow rate of 2 mL min⁻¹. **d** Thickness of the liquid film calculated from gas-liquid flow pattern at various liquid flow rates with a fixed gas flow rate of 2 mL min⁻¹.

affect propane oxidation rate until the supplied $H_2SO_4$ is nearly depleted (i.e., the remaining $H_2SO_4$ concentration at outlet lower than 0.1 M) as suggested by our control experiment (Supplementary Fig. 5). The decrease in liquid flow rate reduces the $H_2SO_4$ feed rate, and thus leading to a lower $H_2SO_4$ concentration at any given point in the reactor barring the inlet. At the lowest liquid flow rate of 0.05 mL min⁻¹, the supplied $H_2SO_4$ was approaching depletion at the outlet of the reactor (Supplementary Fig. 6a), leading to a decrease in the propane oxidation rate (Fig. 2a). The $O_2$ consumption is also increased at low liquid flow rate primarily due to the enhanced Cu dissolution resulting from improved mass transfer and extended residence time of $O_2$ (Supplementary Fig. 6a). This is expected to reduce the propane oxidation rate as $O_2$ is the oxidant for activating $C_3H_8$. However, this contradicts to the improved propane reaction rate at lower liquid flow rate, which is attributable to the positive influence of improved mass transfer on propane activation overweighing the negative effect of the reduced $O_2$ concentration. We note that the flow patterns at the reactor's outlet maintained the characteristics of Taylor flow, with slightly reduced length of the gas bubbles due to the consumption of $O_2$ during the reaction (Supplementary Fig. 7).

Increasing the gas feed at higher flow rates can alleviate the reduction in $O_2$ concentration along the Cu microtube and improve the oxidation of propane. At a fixed liquid flow rate of 0.2 mL min⁻¹, the propane oxidation rate increases with the increase of gas flow rate from 2 to 10 mL min⁻¹ (Fig. 2b), whereas the corresponding $O_2$ conversion rate decreases from near 80% to 18% (Supplementary Fig. 6b).

Notably, the highest production rate of propylene at room temperature was achieved at a liquid flow rate of 0.2 mL min⁻¹ and gas flow rate of 10 mL min⁻¹, reaching a value of 19.57 mmol $m_{Cu}^{-2}$ h⁻¹, this value can be further increased to 87.20 mmol $m_{Cu}^{-2}$ h⁻¹ when increasing the temperature to 40 °C. The production rates of propylene from propane achieved in our Cu microtube reactor are at least one order of magnitude higher than that of many previously reported catalysts which operates at high temperatures[12,13,15,17,20,49,51–57] (Fig. 2c), and significantly higher than that achieved in our previous batch-type reactor[26] (Supplementary Fig. 8). At this reaction condition, the Cu microtube microchannel reactor exhibits stable propylene production with over 92% selectivity in a 12-h experiment (Fig. 2d), demonstrating the stability of this reaction system, which includes both the Taylor flow and the catalyst surface for propane activation. The loss of Cu in a 12-h experiment is unlikely to significantly influence the stability of propane activation as the time for dissolving all the Cu in our Cu microreactor can be estimated to be around 346 h calculated from the Cu dissolution rate (~ 0.133 g h⁻¹) and the mass of Cu microtube (~ 46 g). However, for practical application, the stability of the reactor itself should be taken into account. Designing stable catalysts, in which the specific active sites are immobilized on the wall of the microtube that do not rely on Cu dissolution, represents a promising strategy for improving the stability of the reactor. Increasing the gas flow rate at a fixed liquid flow rare is expected to improve the mass transport (Supplementary Fig. 4)[44,45], however, this is not likely responsible for the enhanced activities as the reaction rate at the liquid flow rate of

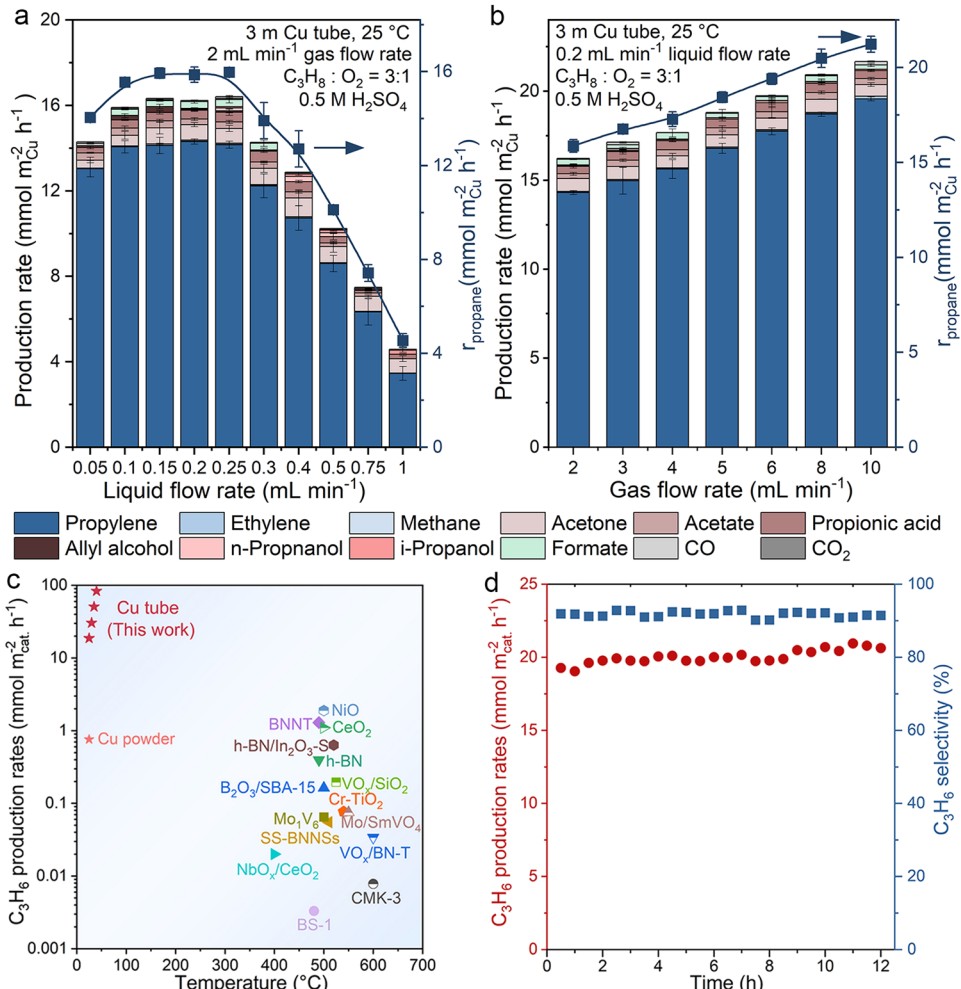

**Fig. 2 | Propane activation in Cu microtube reactor.** Product distributions and propane reaction rates at **a** various liquid flow rates with a fixed gas flow rate of 2 mL min⁻¹ and **b** various gas flow rates with a fixed liquid flow rate of 0.20 mL min⁻¹. The molar ratio of $C_3H_8$ and $O_2$ was fixed to be 3:1. The error bars represent the standard deviation from at least three independent measurements. **c** Comparison of the production rates of propylene verses temperatures between Cu microtube in

this work and state-of-the-art catalysts in literature of ODHP (see Supplementary Table 1 for more details): Cu powder[26], NiO[57], BNNT[17], h-BN[17], CeO₂[52], h-BN/In₂O₃–S[20], VOₓ/SiO₂[12], Cr-TiO₂[51], Mo/SmVO₄[53], Mo₁V₆[54], SS-BNNSs[55], VOₓ/BN-T[49], NbOₓ/CeO₂[13], CMK-3[56] and BS-1[15]. **d** Stability test in a 12-h operation in a Cu microtube reactor at room temperature.

0.2 mL min⁻¹ is not limited by the mass transport even when the gas flow is at the lowest rate of 2 mL min⁻¹ (Fig. 2a). The conversion of propane remains low within the gas flow rate between 2 to 10 mL min⁻¹ (Supplementary Fig. 9). Thus, the enhanced reaction rate can be solely ascribed to the increase of $O_2$ concentration along the Cu microtube.

**Kinetic analysis of propane activation in Cu microtube reactor**
The impact of reactant concentrations, i.e., partial pressure of $O_2$ and $C_3H_8$, on the reaction rate were then investigated at a gas flow rate of 10 mL min⁻¹ and a liquid flow rate of 0.2 ml min⁻¹, owing to the enhanced mass transport and relatively low $O_2$ consumption (Supplementary Fig. 10). The inert Ar gas was introduced to maintain a constant total gas flow rate when the partial pressure of propane or oxygen was adjusted separately. The propane oxidation rate exhibits first-order dependence on the partial pressure of propane when fixing the $O_2$ partial pressure at 0.25 atm (Fig. 3a). Noting that the $O_2$ conversion rates remained remarkably consistent across various propane partial pressures in the study of propane's reaction order (approximately 18%). Consequently, despite its consumption, the impact of $O_2$ on the reactions should be comparably uniform in the analysis of propane's reaction order. The dependence of reaction rate on $O_2$ partial pressure at the fixed propane partial pressure of 0.25 atm

resembles a Langmuir chemisorption model (Fig. 3b). The Langmuir chemisorption behavior of $O_2$ partial pressure is commonly observed in ODHP on boron-based catalysts, in which the $O_2$ is first adsorbed onto boron sites to activate the catalysts[15,17]. In our system, the impact of $O_2$ concentrations on propane activation rate is likely linked to the dissolution of Cu as its rate exhibits a similar trend as that of propane oxidation with respect to varying $O_2$ concentrations (Supplementary Fig. 11a). An almost linear correlation between the propane activation rate and the Cu dissolution rate is revealed in Supplementary Fig. 11b, further implying that the dissolution of Cu may play a significant role in facilitating C-H activation. These mechanistic aspects will be discussed in the following sections. A higher $O_2$ to propane ratio is beneficial for the deeper oxidation of propane, as both increasing the $O_2$ partial pressure or decreasing the propane partial pressure lead to higher selectivities toward oxygenates, such as acetone and propionic acid (Supplementary Fig. 12). The apparent activation energy with a value of 76.21 kJ mol⁻¹ was determined in the temperature range of 298 to 313 K, which is close to the typical values obtained in oxidative dehydrogenation of propane[13,15,50]. A slight increase in the selectivity of propylene was observed at elevated temperature (Supplementary Fig. 13), likely due to the facilitated desorption of propylene from Cu surface[5,58]. High performance combining a selectivity of 94% and a

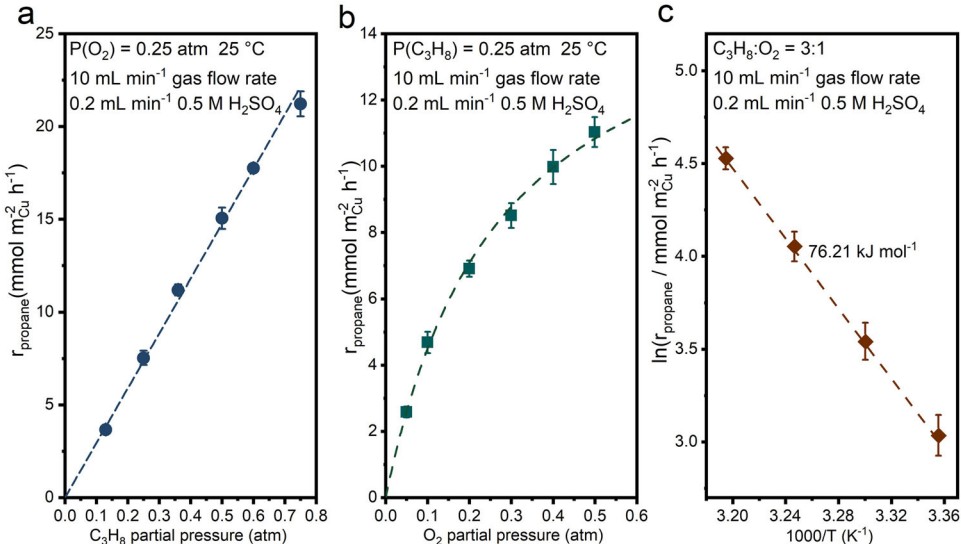

**Fig. 3 | Kinetic analysis of propane activation in Cu microtube reactor.** The dependence of propane oxidation rate on **a** propane partial pressure and **b** oxygen partial pressure. **c** Arrhenius plot of propane activation. The error bars represent the standard deviation from at least three independent measurements.

production rate of 87.20 mmol $m_{Cu}^{-2}$ $h^{-1}$ of propylene was achieved at 313 K. Control experiment using propylene as substrate only produces a small amount of allyl alcohol (Supplementary Fig. 14), suggesting that propylene was not an intermediate in route to oxygenates (except for allyl alcohol) or $CO_2$ in propane activation.

### Computational investigations for mechanistic understanding

To explore the origin of high selectivity of propylene in propane activation, DFT calculations were carried out to explore reaction pathways toward propylene and other minor product formations on Cu surface in acidic environment. Both kinetics and thermodynamics were considered. $Cu_2O$ was selected as the model surface for DFT calculation in this reaction framework, which was justified by the fact that a thin $Cu_2O$ layer readily forms on Cu surfaces upon contact with $O_2$ molecules as evidenced by both experimental and theoretical investigations reported in literature[26,59,60]. The possible role of $Cu^{2+}$ on propane activation can be ruled out by the fact that no products produced when conducting the control experiment that containing 0.2 M $CuSO_4$ and 0.5 M $H_2SO_4$ in the liquid flow using a polypropylene tube rather than Cu microtube as the reactor (Supplementary Fig. 15). As shown in Fig. 4, the initial propane activation can proceed through the cleavage of either primary or secondary C-H bond, leading to the formation of a Cu bound n-propyl or i-propyl and an adjacent surface hydroxyl group (Fig. 4, Supplementary Fig. 16 and 17). This finding is in line with previous studies on ODHP, underscoring the cleavage of both primary and secondary C-H bond to be the key step in activating propane[15,23,24,50]. The initial cleavage of both primary and secondary C-H bonds is exothermic and kinetically facile, with energy barriers of only 0.23 and 0.26 eV, respectively, which can be easily surmounted at ambient conditions. Following this, protons in acidic solutions react with the surface hydroxyl group to form water which desorbs from the surface, leaving behind a positively charged surface with an oxygen vacancy. This vacancy can be readily filled by $O_2$ molecules with $\Delta G$ = -1.77 eV. The resulting positively charged surface exhibits the capability for further C-H cleavage of the adsorbed propyl to form propylene, or C-O bond formation to form propanol. The energy barriers of the further C-H bond cleavage or C-O bond formation along the n-propyl pathway are 0.36 and 1.20 eV, whereas those along the i-propyl pathway are 0.33 and 0.92 eV, respectively. The energy barriers for further C-H bond cleavage on both two pathways are considerably lower than those for C-O bond formation, aligning with the experimental results

that the selectivity of propylene is significantly higher than that of oxygenates during propane activation. Upon the formation of propylene and propanol, the resulting protonated surface reacts with the protons and dissolves the $Cu^{2+}$ ions, exposing fresh Cu surface to react with $O_2$ to form Cu oxide layer for subsequent propane activation. These results suggest a coupling between propane activation and Cu dissolution, explaining the observed consistency between propane activation and Cu dissolution rates across various $O_2$ partial pressures (Fig. 3b and Fig. S7).

## Discussion

In this study, we developed a Cu microtube reactor in conjunction with gas-liquid Taylor flow enabling efficient propylene production via propane activation at room temperature and ambient pressure. The exceptional performance observed can be attributed to significantly improved mass transport, facilitated by a substantial gas-interfacial area and a minimized path for reactant diffusion to the catalyst surface within the gas-liquid Taylor flow. The Taylor flow, obtained by the T-junction microdevice, demonstrated noteworthy stability, promoting the continuous production of propylene with an impressive selectivity exceeding 92% over a duration of at least 12 hours. We further demined key kinetic variables, such as reaction orders and apparent activation energy, enabled by the enhanced mass transfer. DFT calculations reveal that the propane activation process is coupled with a facile Cu dissolution reaction, aligning with observed trends where the dependence of propane activation rate and Cu dissolution rate on $O_2$ concentration marked similarly. Both computational and experimental analyses show a proclivity of this reaction system toward the production of olefins over oxygenates, suggesting the potential for the production of long-chain alkenes, which are important platform chemicals predominantly derived from petroleum. This strategy can be extended to facilitate other multiphase reactions involving reactants with low solubility in a liquid medium. The microfluidic reaction system presents scalability advantages, as it can be easily upscaled using multiple microchannels in parallel with a shared flow. Importantly, our system also mitigates the potential explosion risk associated with flammable gas and oxygen mixture, as the gaseous mixture is segmented into small millimeter-scale portions by plugs of aqueous solution. While recognizing that propane conversion in this system remains modest, effective strategies to overcome this limitation include the development of a packed bed reactor that integrates larger

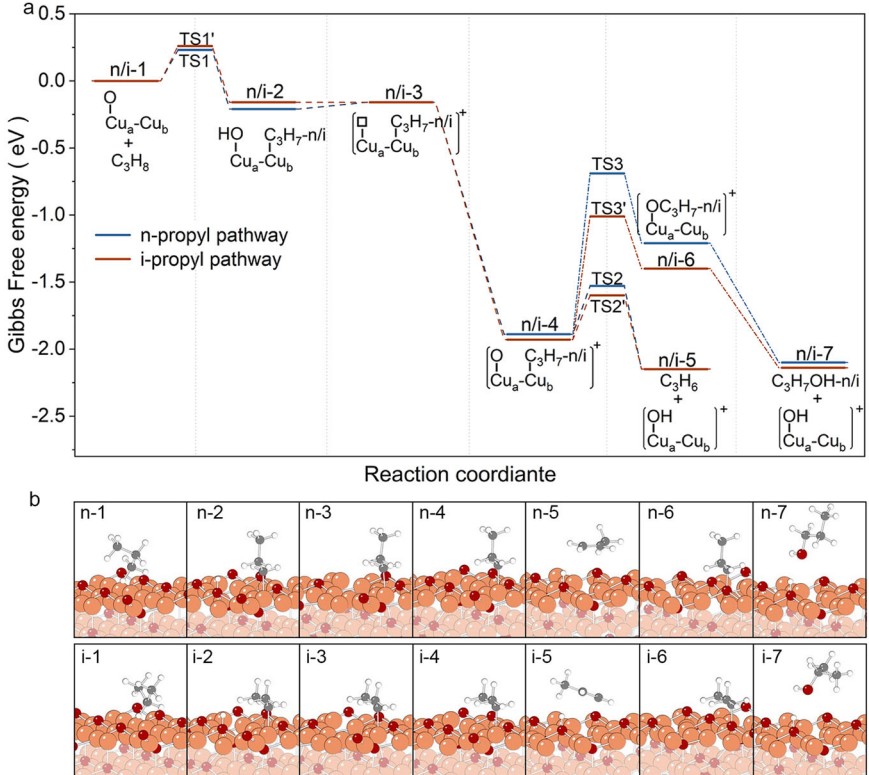

**Fig. 4 | Reaction pathways of propane activation toward propylene and n-propanol/i-propanol formation. a** Free energy diagram where the white square symbol stands for an oxygen vacancy. **b** Structures for each reaction step. Orange, red, grey and white balls stand for Cu, oxygen, carbon and hydrogen atoms, respectively.

Cu catalyst surfaces, in conjunction with microfluidic flow, and operates at higher temperatures. Importantly, future efforts should also focus on the design of stable catalysts featuring specific active sites that do not rely on Cu dissolution. This approach not only aims to reduce Cu consumption during propane activation but also represents a promising avenue for advancing research in this field.

## Methods

### Materials
Cu tube (Zhongxin Metal Materials, 550 μm inner diameter, 99.9%), sulfuric acid (Sigma, ACS reagent, 95.0-98.0%), Cu powder (Sigma-Aldrich, dendritic, <45 μm, 99.7% trace metals basis), propane (Air Liquide, 99.9%), $O_2$ (Air Liquide, 99.999%), Ar (Air Liquide, 99.999%), propylene (Air Liquide, 99.99%). Deionized water in all experiments was prepared from a Millipore system (18.2 MΩ·cm)

### Gas-liquid Taylor flow formation
Commercial Cu microtubes with inner diameter of 550 μm and length of 3 m were used as microchannel reactors for all experiments unless noted otherwise. Two transparent polypropylene tubes with identical diameter were connected to both ends of the Cu microtube for capturing the flow pattern. Prior to entering the Cu microtube, the reactant gases, composed of $C_3H_8$ and $O_2$ with a predetermined ratio, were dispersed with aqueous $H_2SO_4$ solution using a custom-made capillary embedded step T-junction microdevice[40]. The flow rates of $C_3H_8$ and $O_2$ were controlled by mass flow controllers (MKS Instruments Inc.) and calibrated by an ADM flow meter (Agilent Technologies). The flow rate of aqueous $H_2SO_4$ solution was controlled by a syringe pump (Chemyx Inc.) and calibrated by a graduated cylinder. After passing through the Cu microtube, the gas components in the flow were collected using downward displacement of water, while the liquid was collected separately using a vial. The flow pattern was captured using a high speed camera (i-SPEED TR, Olympus) with a fixed frame rate of 2000 fps after the flow system was allow to stabilize for at least 5 minutes. The flow patterns as well as the mass transfer characteristics, i.e., thickness of liquid film and gas-liquid interfacial area, were then determined upon the recorded photos (Supplenmantary Note 1).

### Propane activation in a microchannel reactor
The activation of propane was carried out in the above-mentioned Cu microtube reactor which also served as the catalyst. The Cu microtube was firstly washed by delivering 0.5 M $H_2SO_4$ solutions with a flow rate of 0.5 mL min$^{-1}$ for 10 minutes to remove the native oxides. Before reaction, five minutes was arranged for the gas-liquid system to stabilize. The mixture of $C_3H_8$ and $O_2$ with a molar ratio of 3:1 was used as the gas feed, and 0.5 M $H_2SO_4$ was chosen as the reaction medium. For experiments at various liquid flow rates from 0.05 mL min$^{-1}$ to 1 mL min$^{-1}$, the $C_3H_8 + O_2$ gas flow rate was fixed at 2 mL min$^{-1}$. For experiments at various gas flow rates from 2 mL min$^{-1}$ to 10 mL min$^{-1}$, the liquid flow rate was fixed at 0.2 mL min$^{-1}$. For experiments at various $C_3H_8$ and $O_2$ partial pressures, the total gas flow rate and liquid flow rate were fixed at 10 mL min$^{-1}$ and 0.2 mL min$^{-1}$, respectively. The inert Ar gas was introduced to maintain a constant total gas flow rate when the partial pressure of propane or oxygen was adjusted separately. For $Cu^{2+}$ control experiment, a 3 m-long polypropylene tube rather than Cu microtube was used as the reactor with 0.2 M $CuSO_4$ and 0.5 M $H_2SO_4$ in the liquid flow. The concentration of $Cu^{2+}$ was chosen to match the final concentration of $Cu^{2+}$ after propane activation at 10 mL min$^{-1}$ gas flow rate and 0.2 mL min$^{-1}$ liquid flow rate with $C_3H_8$ to $O_2$ ratio of 3:1. For temperature dependence experiments, Cu microtube was immersed in water with preset temperatures. A polypropylene tube with 3 m length connected to the inlet of Cu microtube was also placed in water in order to sufficiently preheat the gas-liquid

Taylor flow. 1 m length of Cu microtube was used in order to avoid high $O_2$ conversion at elevated temperatures.

## Product quantification

The collected gas components were quantified by gas chromatograph (Agilent 7890B) every 30 minutes. The gas chromatography analyses were averaged during a 70-min experiment. The gas chromatograph was equipped with a ShinCarbon ST Micropacked column and a HayeSep Qcolumn. Ar was used as the carrier gas. $C_3H_6$, $C_3H_8$, $CH_4$, $C_2H_4$, CO, $CO_2$ were quantified using a flame ionization detector (FID) with a methanizer, and $O_2$ was quantified using a thermal conductivity detector (TCD). The response factors for these species were calibrated by analyzing a series of standard gas mixtures.

The liquid components were quantified by using a Bruker AVIII 600 MHz NMR spectrometer. Before sampling, $Cu^{2+}$ ions in the postreaction solutions were firstly removed via electrodeposition to avoid interference on NMR signals due to their paramagnetism[61]. The NMR sample was prepared by mixing 500 μL of the processed solution with 100 μL of $D_2O$ (Sigma-Aldrich, 99.9 atom % D) and 0.05 mM dimethyl sulfoxide (DMSO, Alfa Aesar, ≥99.9%) as internal standard. Acetate anion standard solution (Aladdin, 0.1 mg·mL$^{-1}$) was used to calibrate the concentration of DMSO. The $^1H$ spectrum was measured with water suppression by using the excitation sculpting method.

The production rate of gaseous product $i$ was calculated as the following equation. $Q_G$ is the gas flow at the outlet; $x_i$ is the molar fraction of product $i$ of the gas flow at the outlet determined by GC; $V_m$ represents molar volume of gas at room temperature (24.5 L mol$^{-1}$); $S$ is the inner surface area of Cu microtube.

$$r_i = (Q_G \times x_i)/(V_m \times S) \tag{1}$$

The production rate of liquid product $i$ was calculated as the following equation. $Q_L$ is the gas flow rate and liquid flow rate at the outlet; $c_i$ is the molar concentration of product $i$ in the liquid flow at the outlet, which was determined by NMR; $S$ is the inner surface area of Cu microtube.

$$r_i = (Q_L \times c_i)/S \tag{2}$$

The selectivity of product $i$ is based on carbon atoms and was calculated using the following equation:

$$S_i = \frac{r_i}{r_{total}} = r_i / \left( r_{C_3H_6} + \frac{2}{3}r_{C_2H_4} + \frac{1}{3}r_{CH_4} + r_{acetone} + r_{propionic\ acid} \right.$$
$$\left. + r_{n-propanol} + r_{i-propanol} + r_{allyl\ alcohol} + \frac{2}{3}r_{acetate} + \frac{1}{3}r_{CO} + \frac{1}{3}r_{CO_2} \right) \tag{3}$$

The conversion of propane and $O_2$ can be calculated as follows, where A stands for the amount of substance.

$$X_{C_3H_8} = \frac{A_{C_3H_8}^{in} - A_{C_3H_8}^{out}}{A_{C_3H_8}^{in}} \times 100\% \tag{4}$$

$$X_{O_2} = \frac{A_{O_2}^{in} - A_{O_2}^{out}}{A_{O_2}^{in}} \times 100\% \tag{5}$$

The concentrations of remaining $H_2SO_4$ and formed $Cu^{2+}$ were calculated from the $O_2$ consumption in Cu dissolution based on the stoichiometric reaction as follows:

$$2Cu + O_2 + 4H^+ = 2Cu^{2+} + 2H_2O \tag{6}$$

The $O_2$ consumption in Cu dissolution can be calculated by using total consumption of $O_2$ minus the the amount of $O_2$ consumed in propane activation, which can be calculated from the amount of products according to the reaction equation of $C_3H_8$ and $O_2$ toward corresponding products.

The concentration of $Cu^{2+}$ in the postreaction solutions was also analysed using a UV-Vis spectroscopy (Cary 4000 UV-Vis spectrophotometer). The working curve was obtained by analyzing a series standard $Cu^{2+}$ sample solutions. The measured $Cu^{2+}$ concentration from UV-Vis was verified with the result obtained based on Eq. (6) within a difference of less than 4%.

## Computational details

$Cu_2O$ was selected as the model surface for DFT calculation in this reaction framework, which was justified by the fact that a thin $Cu_2O$ layer readily forms on Cu surfaces upon contact with $O_2$ molecules as evidenced by both experimental and theoretical investigations reported in literature[26,59,60]. We used a slab ($3 \times 3$) consisting of 3 layers of $Cu_2O(111)$ with the bottom layer fixed in its bulk position to simulate the Cu surface. The projector augmented wave (PAW) method was used to calculated the interaction between atomic cores and valence electrons as implemented in Vienna ab initio Simulation Package (VASP)[62–64]. The Perdew-Burke-Ernzerhof (PBE) functional[65] was used to treat the exchange-correlation interaction. The plane-wave cut off was set to 400 eV. The semi-empirical $D_3$ approach was employed to describe London dispersion interactions as implemented in spin-polarized VASP. To avoid interactions between successive slabs due to the periodic boundary conditions, a vacuum of 50 Å was introduced to the supercell. The reciprocal space was sampled by using a Monkhorst-Pack k-point mesh of $4 \times 4 \times 1$[66]. Each calculation is considered as converged if the electronic energy between two self-consistency steps is smaller than $10^{-4}$ eV. We applied the approach proposed by Head-Gordon et al. and Goddard et al[67–69] to simulate the charged surface interacting with the solvent. In this approach, the linear Poisson-Boltzmann implicit solvation model with a Debye screening length of 3.0 Å was used to neutralize the non-zero charge in the simulation cell and to simulate water and electrolyte, allowing for a more realistic description of the electrical double layer.

The free energies of the slab systems were calculated as follows:

$$G = E_{elec}^{solv} + ZPVE + H_{vib} - TS_{vib} \tag{7}$$

where $E_{elec}^{solv}$ is the electronic energy of the system calculated from VASP. We treated all degrees of freedom of the adsorbates as vibrational and neglect the contribution of vibrations of the slab. The finite difference method was used to evaluate the vibrational frequencies ($v$) by calculating the partial Hessian matrix. To avoid unphysically large entropy contributions, unusually low vibrational modes (<50 cm$^{-1}$) were corrected to 50 cm$^{-1}$. The zero-point vibrational energy (ZPVE), vibrational contributions to the internal energy ($H_{vib}$) and entropy ($S_{vib}$) at 298 K were calculated as follows:

$$ZPVE = \sum_{v} \frac{hv}{2} \tag{8}$$

$$H_{vib} = \sum_{v} \frac{hv}{e^{hv/k_B T} - 1} \tag{9}$$

$$S_{vib} = k_B \sum_{v} \left[ \frac{hv}{k_B T(e^{hv/k_B T} - 1)} - \ln\left(1 - e^{-hv/k_B T}\right) \right] \tag{10}$$

The free energies of small molecules were determined as follows:

$$G = E_{elec}^{solv} + ZPVE + \left(\frac{n}{2} + 1\right)k_B T + H_{vib} - T(S_{vib} + S_{trans} + S_{rot}) \tag{11}$$

where n is 5 for linear molecules and 6 for non-linear molecules. ZPVE was calculated as shown above. $H_{vib}$, $S_{vib}$, $S_{trans}$, and $S_{rot}$ were obtained from Jaguar by using the PBE/6-31 G* basis set. The free energy of $H^+$ was estimated from the pKa of $H_3O^+$ and the calculated free energies of $H_3O^+$ and $H_2O$ using the solvation model. Detailed descriptions of this approach has been provided in our previous works[70–74].

The transition state of each reaction was firstly approached using the nudged elastic band (NEB) method[75]. The plane-wave cutoff, functional, smearing parameter, and calculator parameters were the same as those used in slab geometry optimizations. Structures obtained from NEB were employed to generate the input structure and orientation for the dimer calculation[76]. The force of the dimer calculation was converged to <0.1 eV Å$^{-1}$ to locate the saddle point accurately, i.e., the transition state, which were further verified with the vibrational frequency analysis. The reaction free energy is chosen as the free energy barrier when the calculated free energy barrier is smaller than the corresponding reaction free energy or zero.

## Data availability
The data that support the findings of this study are available from the corresponding author upon request. Source data are provided with this paper.

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

## Acknowledgements

C.L., H.Z., W.L., and Q.L. acknowledge the financial support from the State Key Laboratory of Chemical Engineering (no. SKL-ChE-23T02). L.S. and G.L. acknowledge the financial support from the National Natural Science Foundation of China (21991104). B.X. acknowledges the financial support from Beijing National Laboratory for Molecular Sciences.

## Author contributions

C.L., G.L., and Q.L. designed the project. C.L. and W.L. performed alkane activation experiments. C.L. and L.S. conducted flow pattern analysis. H.Z. and M.-J.C. performed DFT calculations. B.X. and G.L. contributed to data analysis and discussion. Q.L. supervised the entire project.

## Competing interests

The authors declare no competing interests.
