## [Peer Review File · Nature Communications]

Efficient conversion of propane in a microchannel reactor at ambient conditionsREVIEWER COMMENTS

Reviewer #1 (Remarks to the Author):

In this manuscript, the authors demonstrated a microchannel reactor for improving propane partial oxidation to propene under room temperature. The reported propene production rate is comparable to those achieved at elevated temperatures/pressure. Additionally, the authors provided mechanistic insights based on the reaction order analysis and DFT calculations. Beyond enhancing propane conversion, this innovative reactor design has implications for improving mass transport in other gas-liquid reaction systems. While this manuscript is well written and can be recommended for publication, the authors are advised to address a few minor concerns:

1. The authors should confirm that the flow pattern remains consistent at the microtube reactor's outlet, especially considering the substantial consumption of reactants like O₂, which might alter the flow pattern.
2. A justification for the selection of the Taylor flow over other flow patterns, such as bubbly flow, is needed.
3. Including an illustrative scheme or photograph of the T-junction would enhance clarity about how the Taylor flow originates. Additionally, a comprehensive schematic of the entire setup would be beneficial for readers.
4. Essential physical characterizations of the Cu tube in use, such as its chemical composition and morphology, should be presented.
5. The significant consumption of reactants, notably O₂, should be incorporated into the reaction order analysis.
6. Considering future industrial applications, the propane conversion rate needs enhancement. The authors should proffer potential strategies for augmenting the propane conversion rate or curtailing Cu consumption.

Furthermore, the following points require correction or clarification:

1. A clear method on how selectivity and production rates were determined is needed.
2. For Supplementary Figure 1(b), the abscissa title should read "gas flow rate", and its caption adjusted to "at various gas flow rates with a constant liquid flow rate".
3. Within the Supplementary Information, both QL and QG must be distinctly specified.

Reviewer #2 (Remarks to the Author):

This study is a continuous work of Nat. Catal. paper (Nature Catalysis volume 6, pages 666–675 (2023)). Herein the authors reported the selective oxidation of propane in a Cu microreactor under mild reaction conditions. The designed microchannel reaction system enabled highly selective and active propylene production at room temperature and ambient pressure with mitigated safety risks. The results are very interesting in terms of reactor design. However, several questions should be well addressed before

publication.

1. The authors have claimed that the dissolution of Cu may be responsible for the C-H activation. What is the relationship between propane conversion vs. Cu dissolution rate?
2. It should be clarified that whether Cu₂O or soluble Cu²⁺ is the working species? This issue is very important regarding to the explanation of reaction mechanism. Because in the theoretical calculation, the authors employed Cu₂O as the model surface, which could be totally wrong.
3. The authors have provided the stability of propane conversion within a 12 h operation in the Cu microtube reactor at room temperature. It seems that the propane conversion kept unchanged. My question is what is the stability of such Cu microtube. According to the dissolution rate, the loss of Cu could be as high as 25.6 gCu/(m²*h) under O₂ partial pressure of 0.2 atm. Therefore, the stability of reactor under long-term operation remains uncertain.

Reviewer #1 (Remarks to the Author):

In this manuscript, the authors demonstrated a microchannel reactor for improving propane partial oxidation to propene under room temperature. The reported propene production rate is comparable to those achieved at elevated temperatures/pressure. Additionally, the authors provided mechanistic insights based on the reaction order analysis and DFT calculations. Beyond enhancing propane conversion, this innovative reactor design has implications for improving mass transport in other gas-liquid reaction systems. While this manuscript is well written and can be recommended for publication, the authors are advised to address a few minor concerns:

We thank Reviewer 1 for the positive appraisal.

1. The authors should confirm that the flow pattern remains consistent at the microtube reactor's outlet, especially considering the substantial consumption of reactants like O₂, which might alter the flow pattern.

Response: The flow patterns at the reactor's outlet maintained the characteristics of Taylor flow, despite the consumption of O₂ during the reaction (Figure R1). It is noted that the consumption of O₂ slightly reduces the length of the gas bubbles.

Figure R1. Comparison of the high-speed camera photographs of the Taylor flow at inlet and outlet with various liquid flow rates (L). The gas flow rate was fixed at 2 mL min⁻¹.

Action: We added the following sentences in the 1st paragraph on page 7:

“We note that the flow patterns at the reactor's outlet maintained the characteristics of Taylor flow, with slightly reduced length of gas bubbles due to the consumption of O₂ during the reaction (Supplementary Figure 7).”

We added the following figure to the Supplementary Information as Supplementary Figure 7:

Supplementary Figure 7. Comparison of the high-speed camera photographs of the Taylor flow at inlet and outlet with various liquid flow rates (L). The gas flow rate was fixed at 2 mL min⁻¹.

2. A justification for the selection of the Taylor flow over other flow patterns, such as bubbly flow, is needed.

Response: The reason for selecting the Taylor flow over other flow patterns is primarily due to the two simultaneously achieved advantages: significantly enhanced the gas-liquid interfacial area and effectively reduced the liquid film thickness, thereby facilitating the molecular diffusion of gaseous reactants. These factors are crucial for improving the mass transport of the gaseous reactants in propane activation. In addition, Taylor flow is characterized for its uniformity, making it suitable for conducting mass transport characteristic analyses. In contrast, other flow patterns such as bubbly flow and annular flow as shown in Figure R2, have limitations: the bubbly flow, while increasing the interfacial area, does not reduce the diffusion path, and the annular flow, with an extremely low liquid to gas ratio, does not provide a sufficient liquid supply necessary for propane activation.

Figure R2. Comparison of flow pattern of (a) Taylor flow, (b) bubbly flow and (c) annular flow.

Action: We added the following sentences in the 2nd paragraph on page 3:

“In addition, Taylor flow is characterized for its uniformity, making it suitable for conducting mass transport characteristic analyses. Other flow patterns such as bubbly flow and annular flow (Supplementary Figure 3), have limitations in propane activation: the bubbly flow, while increasing the interfacial area, does not reduce the diffusion path, and the annular flow, with an extremely low liquid to gas ratio, does not provide a sufficient liquid supply necessary for propane activation.”

We added the following figure to the Supplementary Information as Supplementary Figure 3:

Supplementary Figure 3. Comparison of flow pattern of (a) Taylor flow, (b) bubbly flow and (c) annular flow.

3. Including an illustrative scheme or photograph of the T-junction would enhance clarity about how the Taylor flow originates. Additionally, a comprehensive schematic of the entire setup would be beneficial for readers.

Response: Thanks for the comment. A scheme of the entire setup as well as the T-junction is provided in Figure R3.

Figure R3. Scheme of the experimental setup for propane activation in Cu microchannel reactor.

Action: We added the following figure to the Supplementary Information as Supplementary Figure 2:

Supplementary Figure 2. Scheme of the experimental setup for propane activation in Cu microchannel reactor.

4. Essential physical characterizations of the Cu tube in use, such as its chemical composition and morphology, should be presented.

Response: The SEM images of the Cu microtube, and its elemental mapping by EDS are provided in Figure R4.

Figure R4. Photo and SEM image of Cu microtube. (a) A photo of cross section, (b-d) SEM image (b) with the elemental distribution by EDX for Cu (red) (c) and O (cyan) (d).

Action: We added the following sentences in the 2nd paragraph on page 3:

“The morphology and chemical composition of Cu microtube were characterized using scan electron microscopy (SEM) equipped with energy dispersive spectroscopy (EDS) (Supplementary Figure 1)”

We added the following figure to the Supplementary Information as Supplementary Figure 1:

Supplementary Figure 1. Photo and SEM image of Cu microtube. (a) A photo of cross section, (b-d) SEM image (b) with the elemental distribution by EDX for Cu (red) (c) and O (cyan) (d).

5. The significant consumption of reactants, notably O_2 , should be incorporated into the reaction order analysis.

Response: We agree with the reviewer that O_2 was consumed during the reaction. However, the O_2 conversion rates remained remarkably consistent across various propane partial pressures in the study of propane's reaction order (approximately 18%). Consequently, despite its consumption, the impact of O_2 on the reactions should be comparably uniform in the analysis of propane's reaction order.

Action: We added the following discussion on the influence of O_2 consumption on the reaction order analysis of propane, in the 2nd paragraph on page 8:

“Noting that the O_2 conversion rates remained remarkably consistent across various propane partial pressures in the study of propane's reaction order (approximately 18%). Consequently, despite its consumption, the impact of O_2 on the reactions should be comparably uniform in the analysis of propane's reaction order.”

6. Considering future industrial applications, the propane conversion rate needs enhancement. The authors should proffer potential strategies for augmenting the propane conversion rate or curtailing Cu consumption.

Response: We have now proposed strategies in the revised manuscript for augmenting the

propane conversion rate and curtailing Cu consumption.

Action: We added the following discussion in the 1st paragraph on page 11:

“While recognizing that propane conversion in this system remains modest, effective strategies to overcome this limitation include the development of a packed bed reactor that integrates larger Cu catalyst surfaces, in conjunction with microfluidic flow, and operates at higher temperatures. Importantly, future efforts should also focus on the design of stable catalysts featuring specific active sites that do not rely on Cu dissolution. This approach not only aims to reduce Cu consumption during propane activation but also represents a promising avenue for advancing research in this field.”

Furthermore, the following points require correction or clarification:

1. A clear method on how selectivity and production rates were determined is needed.

Response: The methods on how selectivity and production rates calculated were provided on experimental sections in the revised manuscript.

Action: We added the following calculation details to the Methods:

“The production rates of gas products and liquid products were calculated as following equations, *i* stands for a certain product, n_{gi} is the percentage of *i* product of the gas flow at the outlet and n_{li} is the concentration of *i* product of the liquid flow at the outlet, which were obtained by GC and NMR, respectively. Q_g and Q_l are the gas flow rate and liquid flow rate at the outlet, respectively. V_m represents molar volume of gas at room temperature and *t* stands for 60 minutes per hour. *S* is the inner surface area of Cu microtube.

$$r_{gi} = (Q_g \times n_{gi} \times t) / (V_m \times S) \quad (1)$$

$$r_{li} = (Q_l \times n_{li} \times t) / S \quad (2)$$

The selectivity of *i* product can be calculated using following equation:

$$S_i = (r_{C_3H_6} + \frac{2}{3}r_{C_2H_4} + \frac{1}{3}r_{CH_4} + r_{acetone} + r_{propionic\ acid} + r_{n-propanol} + r_{iso-propanol} + r_{allyl\ alcohol} + \frac{2}{3}r_{acetate} + \frac{1}{3}r_{CO} + \frac{1}{3}r_{CO_2}) \quad (3)$$

The conversion of propane and O₂ can be calculated as follows, where *Q* stands for the amount of substance.

$$X_{C_3H_8} = \frac{Q_{C_3H_8}^{in} - Q_{C_3H_8}^{out}}{Q_{C_3H_8}^{in}} \times 100\% \quad (4)$$

$$X_{O_2} = \frac{Q_{O_2}^{in} - Q_{O_2}^{out}}{Q_{O_2}^{in}} \times 100\% \quad (5)$$

2. For Supplementary Figure 1(b), the abscissa title should read "gas flow rate", and its caption

adjusted to "at various gas flow rates with a constant liquid flow rate".

Response: The abscissa title and caption were revised in the Supplementary Information.

Action: The abscissa title and caption were revised in the Supplementary Information.

Supplementary Figure 4. (a) High-speed camera photographs of the gas-liquid Taylor flow at various gas flow rates in a transparent polypropylene tube with identical diameter as Cu tube. The total liquid flow rate was fixed at 0.2 mL min⁻¹. (b) Gas-interfacial area calculated from gas-liquid flow pattern at various gas flow rates with a fixed liquid flow rate of 2 mL min⁻¹. (c) Thickness of the liquid film calculated from gas-liquid flow pattern at various gas flow rates with a fixed liquid flow rate of 0.2 mL min⁻¹.

3. Within the Supplementary Information, both Q_L and Q_G must be distinctly specified.

Response: The Q_L and Q_G were specified in the Supplementary Information.

Action: We have added the following sentence in the Supplementary Information.
“ Q_G is the gas flow rate and Q_L is the liquid flow rate.”

Reviewer #2 (Remarks to the Author):

This study is a continuous work of Nat. Catal. paper (Nature Catalysis volume 6, pages 666–675 (2023)). Herein the authors reported the selective oxidation of propane in a Cu microreactor under mild reaction conditions. The designed microchannel reaction system enabled highly selective and active propylene production at room temperature and ambient pressure with mitigated safety risks. The results are very interesting in terms of reactor design. However, several questions should be well addressed before publication.

We thank Reviewer 2 for the positive appraisal.

1. The authors have claimed that the dissolution of Cu may be responsible for the C-H activation. What is the relationship between propane conversion vs. Cu dissolution rate?

Response: As presented in Figure R5, there is an almost linear correlation between the propane activation rate and the Cu dissolution rate, implying that the dissolution of Cu may play a significant role in facilitating C-H activation.

Figure R5. The relationship between propane conversion rates and Cu dissolution rates.

Action: We added the following sentences in the 1st paragraph on page 8:

“An almost linear correlation between the propane activation rate and the Cu dissolution rate is revealed in Supplementary Figure 11, further implying that the dissolution of Cu may play a

significant role in facilitating C-H activation.”

We revised the following figures as Supplementary Figure 11:

Supplementary Figure 11. The relationship between (a) Cu dissolution rate and O₂ partial pressure, (b) propane conversion rate and Cu dissolution rate.

2. It should be clarified that whether Cu₂O or soluble Cu²⁺ is the working species? This issue is very important regarding to the explanation of reaction mechanism. Because in the theoretical calculation, the authors employed Cu₂O as the model surface, which could be totally wrong.

Response: We appreciate this comment. To investigate the potential role of Cu²⁺ in propane activation, we conducted a control experiment that containing 0.2 M CuSO₄ / 0.5 M H₂SO₄ in the liquid flow using a polypropylene tube rather than Cu microtube as the reactor. As the results shown in Figure R6, no products were detected, suggesting that Cu²⁺ is not the active species responsible for C-H bond activation. The selection of Cu₂O as the DFT surface model was justified by both experimental and theoretical investigations indicating a thin Cu₂O layer readily forms on Cu surfaces upon contact with O₂ molecules (*Phys Rev Lett* **63**, 386-389 (1989), *Surface Science* **119**, 399-410 (1982), *Nat Catal* **6**, 666-675 (2023)).

Figure R6. Propane activation of using polypropylene tube instead Cu microtube containing 0.2 M $CuSO_4$ and 0.5 M H_2SO_4 in the liquid flow.

Action: We added the following sentences in the 1st paragraph on page 9:

“The possible role of Cu^{2+} on propane activation can be ruled out by the fact that no products produced when conducting the control experiment that containing 0.2 M $CuSO_4$ and 0.5 M H_2SO_4 in the liquid flow using a polypropylene tube rather than Cu microtube as the reactor (Supplementary Figure 15).”

We added the following figure in the Supplementary Information as Supplementary Figure 15:

Supplementary Figure 15. Propane activation of using polypropylene tube instead Cu

microtube containing 0.2 M CuSO₄ and 0.5 M H₂SO₄ in the liquid flow.

We added the following experimental details in the Methods in the revised manuscript:

“For Cu²⁺ control experiment, a 3 m-long polypropylene tube rather than Cu microtube was used as the reactor with 0.2 M CuSO₄ and 0.5 M H₂SO₄ in the liquid flow. The concentration of Cu²⁺ was chosen to match the final concentration of Cu²⁺ after propane activation at 10 mL min⁻¹ gas flow rate and 0.2 mL min⁻¹ liquid flow rate with C₃H₈ to O₂ ratio of 3:1.”

3. The authors have provided the stability of propane conversion within a 12 h operation in the Cu microtube reactor at room temperature. It seems that the propane conversion kept unchanged. My question is what is the stability of such Cu microtube. According to the dissolution rate, the loss of Cu could be as high as 25.6 gCu/(m²*h) under O₂ partial pressure of 0.2 atm. Therefore, the stability of reactor under long-term operation remains uncertain.

Response: The loss of Cu is indeed 25.6 g_{Cu}/(m²*h) under O₂ partial pressure of 0.2 atm. The inner surface area and the mass of our 3 m long Cu microtube was determined to be 5.1837 * 10⁻³ m² and 46 g, respectively. Based on these values, the time for dissolving all the Cu in our Cu microreactor can be estimated to be around 346 h. Thus, the loss of Cu in a 12-h experiment is unlikely to significantly influence the stability of propane activation. We note that main goal of the 12-h experiment was to demonstrate the stability of the reaction system, which includes both the Taylor flow and the catalyst surface for propane activation. However, for practical application, the stability of the reactor itself should be taken into account. Designing stable catalysts, in which the specific active sites are immobilized on the wall of the microtube that do not rely on Cu dissolution, represents a promising strategy for improving the stability of the reactor.

Action: We added following sentences in the 2nd paragraph on page 7:

“At this reaction condition, the Cu microtube microchannel reactor exhibits stable propylene production with over 92% selectivity in a 12-h experiment (Figure 2d), demonstrating the stability of this reaction system, which includes both the Taylor flow and the catalyst surface for propane activation. The loss of Cu in a 12-h experiment is unlikely to significantly influence the stability of propane activation as the time for dissolving all the Cu in our Cu microreactor can be estimated to be around 346 h calculated from the Cu dissolution rate (~ 0.133 g h⁻¹) and the mass of Cu microtube (~ 46 g). However, for practical application, the stability of the reactor itself should be taken into account. Designing stable catalysts, in which the specific active sites are immobilized on the wall of the microtube that do not rely on Cu dissolution, represents a promising strategy for improving the stability of the reactor.”

REVIEWERS' COMMENTS

Reviewer #1 (Remarks to the Author):

The revised paper can be recommended for publication since it has addressed all our concerns.

Reviewer #2 (Remarks to the Author):

The authors have well addressed my concerns. I recommend acceptance in its present form.